# Illness perceptions of occupational hand eczema in German patients based on the common-sense model of self-regulation: A qualitative study

Anna-Sophie Buse [1,2]*, Annika Wilke[1,2], Swen Malte John[1,2], Andreas Hansen[1,2]

1 Department of Dermatology, Environmental Medicine and Health Theory, Institute for Health Research and Education, University of Osnabrück, Osnabrück, Germany, 2 Institute for Interdisciplinary Dermatological Prevention and Rehabilitation (iDerm), University of Osnabrück, Osnabrück, Germany

☯ These authors contributed equally to this work.
* anna-sophie.buse@uni-osnabrueck.de

**Data Availability Statement:** Data cannot be shared beyond individual quotations in the manuscript because participants did not give consent in terms of publication of their full

## Abstract

### Background

Occupational skin diseases (OSD) in the form of hand eczema (HE) are a common work-related disease. Illness perceptions as presented in Leventhal's Common-Sense Model (CSM) are important for patients' self-management of diseases. Understanding these illness perceptions is essential for patient communicating. No quantitative or qualitative studies which investigated subjective illness perceptions in patients with occupational HE utilized the CSM as theoretical framework. The Objective of this study is to investigate illness perceptions of patients with occupational hand eczema (HE) using the CSM.

### Methods

We applied an exploratory qualitative approach and conducted purposive sampling. Thirty-six patients with occupational HE were interviewed using an interview guide based on the dimensions of the CSM, including coherence and emotional representation. All participants participated in a three-week inpatient program at a clinic specialized on occupational dermatology. One interview had to be excluded before analysis, since one participant's diagnosis was retrospectively changed from ICD to tinea and hence did not match the inclusion criteria. Thirty-five interviews were transcribed verbatim and analyzed. Data was analyzed deductively and inductively using qualitative text analysis. MAXQDA 2018 (Verbi, Berlin, Germany), a software for qualitative data analysis, was applied for coding and summarizing of results. All dimensions of the CSM were explored for occupational HE.

### Results

Several sub-categories could be identified. Participants named a variety of causes in different areas (e. g. external irritants and other hazardous factors, psycho-social factors, allergies, having a 'bad immune system' or lifestyle). The great impact of the disease on the

transcript. The interviews contain sensitive patient data. This decision was made in accordance with the ethics committee of the University of Osnabrück. Details on data are available upon request from the corresponding author or by the Institute for Health Research and Education, Department of Dermatology, Environmental Medicine and Health Theory, University of Osnabrück, Am Finkenhügel 7a, 49076 Osnabrück, Germany (email: johnderm@uos.de).

**Funding:** The project 'Mixed-methods study to assess illness perceptions of patients with occupational contact dermatitis of the hands for enhancing patient education and counseling' ['SubjeKt: Mixed-Methods-Studie zur Erfassung subjektiver Krankheitstheorien von Patientinnen und Patienten mit berufsbedingten Handekzemen für die Schulungs- und Beratungspraxis'], project no. ext FF_1436, was funded by the Institution for Statutory Accident Insurance and Prevention in the Health and Welfare Services [Berufsgenossenschaft für Gesundheitsdienst und Wohlfahrtspflege]. Furthermore, the publication of this study was supported by Deutsche Forschungsgemeinschaft (DFG) and Open Access Publishing Fund of Osnabrück University. The funders had no role in study design, data collection and analysis, decision to publish, or preparation of the manuscript.

**Competing interests:** The authors have declared that no competing interests exist.

participants' life is shown by the wide range of consequences reported, affecting all areas of life (i. e. psychological, physical, occupational, private). Considering coherence, an ambivalence between comprehensibility and non-comprehensibility of the disease is apparent.

## Discussion

The complexity of illness perceptions presented in this paper is relevant for those involved in HE patient education and counseling, e. g, health educators, dermatologists, and, occupational physicians. Future research might further investigate specific aspects of illness perceptions in patients with occupational HE, especially considering the complexity of coherence and overlapping dimensions (i. e. emotional representation and psychological consequences).

## Introduction

Occupational skin diseases (OSD), especially hand eczema (HE), are one of the most common work-related diseases in Europe [1] and play an important role worldwide, especially in industrialized countries [2–8]. For instance, hairdressers, healthcare workers and metal workers are at particularly high risk of developing occupational HE due to skin exposure to irritants and allergens [9–16]. Most occupational HE occur as contact dermatitis (CD) [17], predominantly as irritant contact dermatitis (ICD), caused by skin exposure to irritants (e. g., cleaning agents, detergents) [18, 19], and/or allergic contact dermatitis (ACD) [20, 21], resulting from skin contact to an allergen the individual patient is sensitized to (e. g., hair dyes, fragrances, rubber accelerators) [22, 23]. Depending on one's genetic disposition, atopic dermatitis (AD) and psoriasis are further risk factors for developing occupational dermatoses and, in case of AD, often occur in combination with ICD [17]. Thus, diagnoses are frequently combined in a single patient (e. g., AD and ICD) [24]. The symptoms of ICD, ACD and AD are similar (e. g., itchiness, vesicles, redness). In regard to treating ICD or ACD, it is important to eliminate irritants and/or allergens causing the condition to prevent the disease from becoming chronic [18]. Besides its socio-economic impacts, both on a patient's life and society [21, 25], occupational HE are associated with a high burden of disease and a reduced quality of life [26–30].

The individual behavior (e. g., use of skin protection products) of patients with occupational HE influences the course of the disease and the ability to work [31]. Training and counseling these patients is vital to initiate and support behavior change processes [32]. To provide tailored patient education and counseling interventions, addressing illness perceptions in training and counseling is important [33, 34]. However, these illness perceptions need to be identified and understood in depth, considering all relevant facets. Until now, only few quantitative or qualitative studies have investigated perceptions of patients with eczematous skin diseases [35–43]. Only two studies specifically focused on OSDs [35, 37] and four have used a qualitative approach [35–37, 42]. None of these studies utilized a specific theoretical framework to investigate illness perceptions. Our paper uses the widely used common sense model as theoretical framework (see below) to investigate illness perceptions. Furthermore, the qualitative approach of this paper allows to understand and get a thorough insight into the patients' perspectives [44, 45]. This also includes exploring the patients' wording in regard to their illness perceptions.

## Theoretical framework

Leventhal's CSM framework is commonly used to understand and describe the processes involved in initiating and maintaining behaviors to manage illness threats [46]. According to this model, patients form their own mental representation of their illness which comprise individual illness perceptions. Patients utilize these illness perceptions to respond to and manage an existing or potential future health threat. Illness perceptions derive from the patients' personal experiences (e. g., prior illnesses), the interaction with their socio-cultural environment (e. g., doctors or colleagues) as well as their existing knowledge and beliefs about the disease [34]. According to Leventhal, illness perceptions are characterized by five dimensions: identity, timeline, consequences, cause and control. Identity refers to the name or label of a disease (e. g., eczema, rash) as well as the perceived symptoms [46]. Timeline describes the duration and course (e. g., constant aggravation, cyclical) of the illness. Consequences comprise individual experiences and anticipated beliefs about the effects (e. g., physical or social) of the illness on the patient's life. Cause reflects the patient's ideas on the etiology/triggering factors of the disease. Control or controllability encompasses both medical and personal treatment measures and their effectiveness.

In qualitative research, several studies have used the CSM as theoretical framework to investigate illness perceptions for different diseases (e. g., [47–50]). In quantitative studies, the Illness Perception Questionnaire (IPQ) [51] and its revised version (IPQ-R) [52] are widely used instruments for assessing illness representations. Both questionnaires are based on the dimensions of the CSM. However, the IPQ-R added further dimensions, i. e., coherence and emotional representation [52]: The CSM states that patients develop emotional representations during the process of self-regulation [34]. Emotional representation is considered a determining factor of the evaluation of health and illness [46, 53]. Coherence is understood as the patients' perceived ability to understand their illness. To the best of our knowledge, only two quantitative studies on illness perceptions of patients with hand eczema assess coherence as a dimension of illness perceptions of eczematous skin diseases so far [34, 35].

## Objective

The objective of this study is to utilize the dimensions of the CSM–including coherence and emotional representation–to qualitatively explore illness perceptions of occupational hand eczema.

## Methods

This study was approved by the ethics committee of the Osnabrück University (no. 13/2020). The report follows the Standards for Reporting Qualitative Research (SRQR) [54]. A completed SRQR checklist is attached in S1 Appendix.

## Research design

We conducted an exploratory qualitative study in a healthcare center specialized on occupational dermatology in Osnabrück, Germany, to explore the complex illness perceptions on occupational HE considering the CSM domains identity, timeline, cause, control, consequences, coherence and emotional representation. This qualitative approach enables assessing detailed information about how individual patients cope and understand the complex disease pattern of occupational HE.

## Recruitment and sample

One to three patients out of approximately eight patients which are weekly admitted to a three-week inpatient tertiary prevention of HE program [31], were invited by two researchers (AB and AH) to take part in the study at the end of a routine patient education seminar which takes place at the beginning of the second week of the inpatient stay. Potential participants were verbally informed about the study and the overall interviewing process. Participation was voluntary. Participants were informed that non-participation and withdrawing consent would not result in any disadvantages. Afterwards, written information on the study and consent forms were handed out to participants who verbally declared interest in taking part in the study. After a sufficient amount of time (approx. two to three days), researchers (AB and AH) obtained the participant's written consent. If necessary, participants could ask questions about the consent form at any time. Afterwards, interviewer (AB or AH) and participant agreed on a meeting for conducting the interview.

The purposive sampling aimed at including a typical composition of the clinic's inpatients (see introduction). Inclusion criteria were written consent, sufficient German oral language skills to participate in an interview and being over 18 years of age (German legal age). Patients aged 22 to 63 with occupational HE were recruited based on a sampling plan which considered a balance between age (three age groups: 18–35, 36–50 and over 50), gender (18 men and 18 women), occupation and form of HE. Especially metal workers and health professionals were included (see introduction). However, also salesmen and -women were recruited since these occupations are also common amongst the group of patients of part of the program mentioned above. The participants mostly suffered from ICD and/or ACD, partly combined with AD or psoriasis. The diagnosis was made by experienced dermatologists and reported in written form. Patients diagnosed with tinea, no eczematous skin disease on their hands or exclusively non-occupational skin diseases were excluded from the study.

## Interview guide

The interview guide (see S2 Appendix) comprises of 7 open-ended questions and is based on the dimensions of the CSM including coherence and emotional representations. The optional sub-questions of the interview guide were used to generate more detailed answers if necessary. Interviews began with an open-ended question related to interviewee's general experiences with their skin to give participants space to tell their stories [55].

## Data collection

Thirty-six interviews were conducted between May and October 2021 in a separate room of the clinic. Interviews lasted from 6 minutes 12 seconds to 41 minutes 37 seconds. The average interview length was approx. 16 minutes. Each interview was conducted by one of the researchers (AB or AH). In most weeks during the period of data collection, one to three interviews were completed. In some weeks, no patient was eligible for the study or interested in participation. Both researchers (AB and AH) are experienced in qualitative research and work as health educators at the healthcare center. Using qualitative interviewing techniques, participants were encouraged to describe their own experiences in detail. Due to SARS-CoV-2-related and clinic-internal hygiene guidelines, both interviewee and interviewer wore a face mask and maintained physical distance during the interview process. One interview was excluded afterwards since the participant's diagnosis was retrospectively changed from ICD to tinea and hence did not match the inclusion criteria.

## Data analysis

All interviews were audio recorded with a digital recording device (Olympus DM-770, Olympus, Hamburg, Germany) and transcribed by a professional transcription service (abtipper.de, Digitalmeister GmbH, Hannover, Germany). As explained above, thirty-five transcripts were considered for further analysis. These transcripts were checked and verified by two researchers (AB and AH). Both researchers analyzed the data using qualitative text analysis [56]. MAXQDA 2018 (Verbi, Berlin, Germany) was used for coding and summarizing results. At first, seven deductive main categories based on the illness perception dimensions of the CSM were used for 'a priori' categorization of participants' responses. Afterwards, an inductive approach was used to build sub-categories to specify the different facets of the dimensions reflecting the complexity of occupational HE. To ensure intercoder agreement [56] four interviews were coded independently by both researchers (AB and AH) to identify and eliminate possible coding discrepancies. This process was performed during the deductive as well as the inductive analysis of data. Afterwards, coding of all other interviews was done by the two researchers separately. During and after analysis, the category system (see S3 Appendix) as well as potential ambiguities in the coding process were discussed in regular coding meetings between AB and AH as well as in a qualitative research group to eliminate ambiguities. Summarizing and paraphrasing of results were done by one researcher (AB). The example quotes presented in this paper were translated into English after analysis (AB). All example quotes used in this paper and the original German version of these quotes are also listed in S4 Appendix.

## Results

Thirty-five transcripts were considered for further analysis. The following presentation of the results is based on the category system (see S3 Appendix) used for coding. Due to the quantity of results, only the most striking categories of the category system are presented in this paper.

### Perception of occupational hand eczema

The results regarding the participants' perceptions of occupational hand eczema are based on the five dimensions of the CSM plus coherence and emotional representations. These seven dimensions are outlined in the following:

a. Perceived causes

b. Perception of the timeline

c. Label of the symptoms and illness

d. Perception of controllability

e. Perception of consequences

f. Perceived coherence

g. Emotional representation

**Perceived causes.** Many interviewees reported external irritants and other hazardous influences as triggering factors of hand eczema which are listed in Table 1. Due to the great number of external irritants, main terms (e. g., 'hand washing') were generated to group the factors mentioned. In addition to the irritants and other hazardous factors shown in Table 1, interviewees also named psycho-social factors such as stress (e. g., due to changed working conditions or time pressure), having troubling thoughts, social problems and others (e. g., feeling 'unsettled',

**Table 1. External irritants and other hazardous factors as perceived causes for occupational HE.**

| Irritants | Sample quotes |
|---|---|
| Hand washing:<br>• frequency<br>• doing it wrong<br>• using wrong products | "But then washing isn't good for me either, because of the skin drying out and the pH and everything." (B19) |
| Gloves:<br>• not using protective gloves<br>• long wearing times<br>• double gloving<br>• wet hands in gloves<br>• certain glove materials | "The amount of glove wearing, I think, has become more than it used to be. Yeah, you just put on more. There's no way to avoid it. Sometimes even two pairs on top of each other. So, that's a greater burden for the skin, without question, because the intervals between wearing gloves are also getting shorter and shorter." (B13) |
| Working substances and materials:<br>• water<br>• disinfectants<br>• cleaners<br>• groceries<br>• metal or wood<br>• adhesives | "Oil, grease, cooling lubricant, dielectric, all combined on the skin. And then another cleaner, blowing something off somewhere and then rubbing the cleaner over your hand, rag over it, and so on. Oh, that burns. Well, go wash your hands." (B4) |
| Working environment:<br>• dirty objects (e. g., tools)<br>• dust<br>• air pressure or bad air quality | "Of course, you also have working activities where you touch something, like I said, if you're working in a workshop, there's dirt." (B16) |
| Mechanic work | "The problem was mainly the skin on the joints, that it was torn because of the mechanical problems I still have at work, that I have to do a lot of lifting or twisting or gripping." (B18) |
| Ointment/creams:<br>• wrong product<br>• too many products<br>• cortisone | "After all, you're equipped with thousands of creams. And I don't think that's the right thing for the skin either." (B11) |
| Weather or temperature:<br>• frost<br>• wetness<br>• summer (sweaty hands) vs. winter (dry hands) | "The cold, I guess. Winter, cold, definitely. And [. . .] in the summer, the sun. Also on vacation, it's obvious, the skin also gets more sun." (B28) |
| Sweat | "I think sweating is the main point for me. Increased through all measures now, through Corona, of course. Because, not only sweating on the hands, but on the whole body." (B13) |
| Accumulation of risk factors:<br>• frequent contact to irritants<br>• sometimes several risk factors at once | "Yeah, it's this agglomeration of a lot of things, you know? Wearing gloves, disinfection of hands, also constantly- When I'm wearing gloves, I sweat immediately. And disinfections. My hands are constantly wet. Again, and again, more and more on top." (B20) |
| 'Suspicious substances' | • „As if the truck has been in some construction site where there is something quite strange on the ground where nobody knows what it is which is not traceable. [. . .] It's not just the feeling of sand in the eyes, but something else." (B1) |
| Private sector:<br>• domestic work<br>• arranging furniture<br>• chlorinated pools<br>• dirt (horse riding, fishing) | • "I have a horse. I ride. I always have to make sure that I wear gloves so that the dirt doesn't get on my hands." (B30) |
| Others:<br>• new/unknown working material<br>• itching | "Basically, that was this job-, this change to a, to a new job with a new employer with, with completely new components, new materials that I had in my hand all of a sudden." (B21) |

private problems, anxiety, no time for self-care). Allergies, either being allergic to a specific substance (e. g., creams, gloves or working substances) or a rather abstract concept of 'a general tendency to allergies', as well as genetic factors (e. g., atopic dermatitis, psoriasis, sensitive skin in comparison to 'normal' skin, general genetic predisposition) were stated, too.

Other perceived causes were lifestyle (especially diet or consuming certain groceries, e. g., 'exotic fruits'), a bad or 'aged' immune system, but also aspects considering behavior (e. g., being reckless against one's better judgement).

**Perception of timeline.**  Participants tended to report the onset and evolution of their disease over a period of time by describing the changing of symptoms. The perceived duration of illness varies from four months up to several decades. In general, interviewees stated a cyclical or periodic course of the illness (intervals, phases, episodes, 'like a hormone cycle', 'comes back again and again', rhythm):

"And then it works out for a while. And then there is a relapse, so to speak, and the hands get worse again." (B1)

The length of a phase varies between several days, weeks, or sometimes years. The severity of hand eczema changes from phase to phase, depending on different circumstances: Being on sick or parental leave, being on holiday, weekends, summer season, support and good atmosphere at workplace as well as performing less skin-irritating work were mentioned as circumstances under which the severity decreased. Returning to work after sick leave or being on holiday and winter season (cold and wet weather) were reported as circumstances which lead to aggravation of the eczema. Some interviewees stated that they could not specify under which circumstances the severity of their hand eczema changed. Interviewees reported that they recognized similar symptoms (e. g., vesicles) at the beginning of each phase. Some interviewees mentioned eczema would 'come and go' with phases of complete healing, others stated that it gets better in some phases but does not heal completely. Besides this cyclical course, a few interviewees also stated either a constant aggravation or improvement over the whole course of the illness.

**Label of symptoms and illness.**  The participants reported a great variety of different symptoms experienced which are presented in Table 2. Due to the great number of symptoms, main terms (e. g., 'vesicles') were generated to group the symptoms named. Furthermore, participants used different synonyms to describe hand eczema, e. g., broken skin/hands, skin problems, the (skin) thing, (skin) disease, (skin) eczema, inflammation, skin change, contact dermatitis, rash or bad skin.

**Perception of controllability.**  Participants mentioned different measures adopted either by themselves or healthcare professionals (especially dermatologists) to gain control over their HE. A special focus lay on the usage of creams/ointments. Measures mentioned are categorized in Table 3. Some participants elaborated measures which could be conducted hypothetically in the future or under different (working) conditions. Some participants stated self-harming procedures as both conscious or subconscious measures for relieve, e. g., opening vesicles (with a needle), scratching, picking/nibbling, rubbing, using hot water or scratching until skin bleeds (pain was considered more bearable than itchiness). Some interviewees reported about the perceived effectiveness of certain measures: Either measures were successful (or are considered as being successful in future), moderately/less and less successful (also over a certain period of time, accompanied by a perceived loss of controllability) or not successful at all:

*"They [authors note: doctors] were quite strict. I wasn't used to that, but it was probably the right way to go. That helped a lot. It had a quick effect." (B13)*

*"Then you pay much more attention to it, I also wear the gloves. I have all that, but still because of all these gloves and super great paying attention, it still happens, all the time." (B6)*

Another aspect mentioned by some participants is that some are able to assess the state of their HE:

**Table 2. Perceived symptoms of occupational HE.**

| Symptom(s) | Sample quotes |
| --- | --- |
| vesicles:<br>• itchy<br>• small/big<br>• many<br>• filled with liquid<br>• dried out<br>• like pimples/pox | "Within one or two hours, there were small blisters that itched a lot. And then it just went pretty quickly, it wasn't bearable, which is why I then went to see a doctor." (B1) |
| rhagades/cracks:<br>• deep<br>• skin rips/pops open | "Until they cracked, rhagades, bloody, the desquamation." (B3) |
| peeling of skin:<br>• hangs down in shreds<br>• like after a sun burn | "My hands, they actually looked like- I could peel off a piece of skin just like that every morning. It was hanging down in shreds." (B4) |
| calluses:<br>• thick<br>• like bark | "(. . .) and because of the cracks there were open wounds and after that there was only callus and even if you had a pen in your hand and wanted to write something, it burst open again." (B10) |
| deformation of nails:<br>• wavelike<br>• nails fall off | "The nails are just not the way you want them to be. They're deformed. But they've been that way for a while." (B13) |
| fluids:<br>• oozing<br>• sticky fluids<br>• pus | "With pox, itching, festering, sticky fluid or- well not festering but rather sticky fluid." (B19) |
| general appearance:<br>• bad looks<br>• not nice<br>• hands are disfigured/open<br>• wound<br>• red<br>• scaling | "(. . .) it was really so bad one morning so I got up, and it was fiery red, open" (B2) |
| pruritus, pain:<br>• at night<br>• unbearable<br>itch attacks | "It's just annoying, the itching above all things." (B22) |
| Others:<br>• rough<br>• bloody<br>• raw<br>• swelling<br>• dry<br>• like wax<br>• scars<br>• porous<br>• heat or inflammation 'inside'<br>• thin and sensitive skin | "always with swelling, redness, really thick" (B2).<br>"The surface of the skin was all nicely healed again. It was intact again. But in the depth you still saw these infection-, uh inflammations." (B3)<br>"So there are real lacerations, until the blood comes, so to speak." (B8) |

*"And I actually thought that the hands looked quite good, they were a little red, a little over-heated, but I'm used to something worse." (B20)*

**Perception of consequences.** Participants named various consequences resulting from their illness. Foremost, interviewees focused on psychological, occupational, physical and private consequences. These are listed in Table 4. Few participants also mentioned financial consequences (e. g., expensive products, not being able to pay off mortgage in case of job change, sick benefits much lower than normal wage) or no consequences at all. Many participants

**Table 3. Measures adopted by participants and physicians to control occupational HE.**

| Measure | Examples |
|---|---|
| 'Medicine' or treatments | Creams/ointment/emollients:<br>• adjectives used: special, several, different, expensive, protective, oily, urea<br>• 'mixed' by dermatologist or pharmacist<br>• frequency: often, a lot, 'like a maniac', after work, after showering, not hesitating to use it<br>• apply thickly (even when skin looks alright) or varying amounts according to state of skin<br>• giving the skin 'a treat', in combination with cotton liners at night<br>• having creams handy in flat (i. e. in different rooms), at workplace and in backpacks/bags<br>• pay attention on quality, trying different drug store products |
| | Example quote:<br>"(. . .) lots of creams, different strengths of cortisone, depending on the condition, how extreme it is at that point." (B20) |
| | Others:<br>• baths (e. g., black tea baths, saltwater baths, paraffin wax bath)<br>• light therapy (e. g., PUVA therapy)<br>• adjusting doses (either by physician or oneself), reducing the amount of creams used<br>• taking pills<br>• cortisone<br>• antibiotics |
| Reducing the number of skin-irritating factors | • sick leave (also several weeks/months)<br>• delegating (also at home, e. g., wet work) or rearranging tasks (e. g., to avoid wearing single use gloves), avoiding skin-irritating tasks (both at home and at work)<br>• using skin protection products at work<br>• washing one's hair with gloves<br>• 'avoiding stuff' in general, stop using disinfectants, less hand washing (only when necessary), avoiding substances one is reacting to, stop stroking cats, 'only doing what you have to do'<br>• overall 'adjusting' of everyday life |
| | Example quote:<br>"I talked to my wife about it like that, that I just leave out all these wet work at home." (B24) |
| Gloves | • 'extra special', many different, rubber<br>• when cutting vegetables at home/performing 'certain' tasks<br>• cotton liners<br>• receiving gloves from employer<br>• reducing wearing times |
| | Example quote:<br>"I wear gloves, but I don't wear them too long either, so that I don't sweat in them and so on." (B32) |
| Measures concerning physicians or insurance | • (regularly) consult a physician (general practitioners, dermatologist, occupational physician, otolaryngologist physician), those who are familiar with one's case, consult an 'expert' (on skin diseases)<br>• change physician when not satisfied<br>• report to the employer's liability insurance association (Germany)<br>• rehabilitation<br>• allergy testing<br>• taking skin samples<br>• following treatment plan |
| | • Example quote:<br>"I'm at the dermatologist quite often. So, every three, four weeks, I have appointments to check my skin." (B9) |

(*Continued*)

**Table 3.** (Continued)

| Measure | Examples |
|---|---|
| Others | • cooling/distracting oneself (especially when itchy)/ trying not to itch<br>• less stress, more time for skin<br>• protection plan/concept for skin protection at work<br>• plasters on rhagades<br>• protective equipment<br>• seminars on skin protection<br>• omit certain groceries/products (e. g., shampoos, exotic fruits, nuts)<br>• skin-friendly cleaning product<br>• avoid substances to find a cause<br>• pay more attention to the skin |

expressed their personal way of dealing with the disease which could be understood as a personal 'overall' consequence:

*"Yeah, you, it's just, you learn to live with it, eventually. Yeah, you learn to live with it." (B25)*

Some interviewees said that they would go to work if one is 'able to get up' or do leisure activities even if it is painful. Some interviewees concluded that one 'cannot change it' and–hence–has to accept, adapt to, and learn to live with the disease to continue normal everyday life and cope with it (also with protective measures). One should be optimistic and stay content instead of worrying about it too much. Furthermore, one must learn to take care of and speak for oneself. However, reducing social contacts, hiding hands, withdrawing from public life and not talking to others about it (especially superiors at work) were also mentioned as strategies. Some described the disease as burden which changed life quality for worse.

**Perceived coherence.** Participants stated different aspects of coherence: First, a great number or interviewees discussed the (in)comprehensibility of the emergence or course of the illness (see Table 5). It was noted that skin diseases are complex which is why it is a long process to understand them. Some talked about the overall sense of coherence, considering the presence of necessary resources (e. g., finding and avoiding a certain substance or a new drug) which would eventually ensure that the disease is cured or at least manageable. In contrast, the lack of those resources (e. g., there is no 'miracle drug') leads to a great concern about whether the disease would ever be manageable or turn into something 'even worse':

*"Hopefully it will go away or hopefully it's not incurable or something like that." (B12)*

Furthermore, a few participants described the emergence and course of their illness as fateful (i. e., in the sense of good/bad luck) or a process which is beyond their or other people's control. A few participants considered it unfair that they suffer from the illness (e. g., as colleagues do not have eczema, even though these work with irritant substances, too).

**Emotional representation.** Participants named a variety of mainly bad emotions associated with the illness. These emotions occur depending on the respective situation (e. g., social context with strangers/own children or working context) and comprise shame, nervousness/tension, being irritably, anger, sadness, disgust, frustration and panic. Only one participant described a feeling of happiness, when recognizing an improvement of the skin.

Example quote: *"But I've tried to hide it as well as possible from my children. At least from the little ones. Because he has already come and said 'Dad, your hand looks disgusting.'" (B15)*

**Table 4. Perceived psychological, physical, occupational and private consequences of occupational HE.**

| Sub-Domain | Examples |
|---|---|
| Psychological | • thinking/worrying about it a lot/everything is about the hands, stress<br>• pressure at work<br>• 'pulls you down', feeling bad, crying, depressive mood 'just because of the hands', not content/discontent, sad<br>• burden, limitations in everyday life, not able to carry out everyday activities<br>• mockery, feelings of shame (e. g., in intimate interactions), feeling other's glances, not masculine<br>• feeling excluded, reserved, not going out when skin is itchy<br>• bad life quality, incisions in well-being, unpleasant, sleep deprived<br>• 'behaving differently than usual', nervous, oblivious, inattentive, unbalanced<br>• feeling handicapped, helpless<br>• annoyed by other people's comments (e. g., 'Is it scabies?') and protective measures<br>• loss of self-confidence<br>• not in the mood to do something nice/to enjoy, nothing is motivating<br>• uncertainty, frustration, not knowing where it comes from, despair, disappointment |
| | Example quote:<br>"And when I go shopping, for example, I also started wearing cotton gloves. But I've also had to listen to people, also acquaintances, saying: 'Oh, Michael Jackson.' Stupid comments. And yeah, and then, of course, there's also a bit of shame." (B7) |
| Physical | • hands do not do what they are supposed to<br>• moving differently<br>• open skin, changed skin milieu, skin is more sensitive than it used to be, wounds take longer to heal, scars<br>• itchiness, pain (one cannot use hands normally), burning<br>• suffering from side effects of medicine/pills (are "like a bomb", one needs to relax after taking a tablet)<br>• no bending of fingers/making a fist, skin is tense<br>• no fingerprints<br>• stiff, like leather<br>• skin gets stuck (e. g., to fabrics)<br>• pressure sensitivity<br>• not being able to touch anything, to perceive/feel anything with fingers |
| | Example quote:<br>"But if I say, 'I'll use hand sanitizer.' you'll hear me screaming in pain all the way down the street." (B19) |
| Occupational | • feeling bad about colleagues as they have to overtake tasks<br>• quitting apprenticeship/other jobs in the past because of disease<br>• worried about losing job/being replaced, questions about future career path, thinking about changing jobs (e. g., working in an office), 'getting through the last years of work before retiring'<br>• consulting doctor's during working time and sick leave (probably also in the future): no durable solution, feeling unreliable<br>• not being able to conduct certain tasks or to conduct them differently, esp. when wet/dirty/fine motor activities/tools fall down, slower work pace, less (direct) contact to patients/customers<br>• working with protective gloves, using creams<br>• moral conflict: using disinfectants means pain, not using it is against hygiene rules, using plasters, even though forbidden, also: broken skin cannot be disinfected<br>• questioning if current job is still suitable in context of skin disease<br>• reorganize working procedures/job rotation to avoid skin irritating tasks<br>• glances and comments of colleagues/patients/customers, tips on skin protection are not helpful<br>• socially excluded from activities (e. g., at breakfast table at workplace) |
| | Example quote:<br>"When the hands are really open, you also get the feeling at some point that you are unreliable, because you just have to call in sick when it's really bad. (. . .) Then I'm just always afraid that people will think that maybe I'm not reliable or that it could be used against me." (B2)<br>"Then the others started too, the work colleagues. And they were sitting in a breakfast room and it wasn't nice. They said, 'Sit somewhere else. This doesn't look nice.' I say, 'But I can't do anything about it. It's just the way it is.' Then they said, 'Then put something on.' And then I started wearing gloves in the breakfast room." (B34) |

(*Continued*)

**Table 4.** (Continued)

| Sub-Domain | Examples |
|---|---|
| Private | • hiding hands from kids, being distant to partner and family (preferring to touch with cotton liners instead of rough/bloody hands)<br>• not being able to do leisure sports, making a sandwich, hold a glass of water properly, no social events to protect hands (becoming an outsider), not touching animals/pets, one has to choose between activities<br>• friends' glances and comments, family and friends recognize that you / your skin is not alright<br>• everyday tasks are challenging (e. g., cutting vegetables)<br>• always having creams and gloves handy<br>• using gloves (e. g., when washing hair, walking the dog)<br>• asking friends, family, partner for help even if it 'just' a skin disease (e. g., gardening, washing dishes), not able to help around the house<br>• talk to friends/family to find solutions<br>• not being able to do something rashly, everything has to be thought through but not always in the mood to conduct protective measures<br>• not wearing what you like (clothes, jewelry)<br>• blood on bedclothes |
| | Example quote:<br>"I've just become a grandmother. It's just a bit strange when you're like, 'Do I touch this child now or not?' And when it comes to my partner, I sometimes have the feeling- Do I stroke my husband or will he say, 'Take your rough hands off me?'" (B5) |

**Table 5.** Aspects mentioned in context of incomprehensibility and comprehensibility of occupational HE.

| Dimension | Examples |
|---|---|
| **Incomprehensibility** | • wondering about the cause of the illness, nothing 'special' about everyday work which could cause/suddenly led to the emergence/worsening of a skin disease<br>• wondering why there were no problems in the past (e. g., usage of skin irritating substances in the past but not anymore) or trying to find reasons for why it is occurring now (e. g., changes in the body)<br>• not accepting the diagnosis<br>• wondering if it is an occupational or genetic cause/why it is so severe/got worse<br>• not being able to explain coherencies ('I am not a scientist/doctor')<br>• wondering why it is located on hands and nowhere else<br>• need for a definition of what is going on (e. g., positive patch test result)<br>• understanding of disease processes in the skin but not why they happen<br>• never thought about whether one understands the disease or not<br>• in the past, one's own body was considered as 'not vulnerable' (e. g., taking off gloves when skin is considered healthy)<br>• believe that someone intentionally withholds information (e. g., ingredients of tablet/cream or substances at work, no questioning of medical diagnoses) |
| | Example quote<br>"Actually, yes. Because I also don't know-, I don't know what it is. Then I can also not understand so, wha- or why or why. Or whatever triggered it that the hands are so scaly." (B28) |
| **Comprehensibility** | • understanding the disease after a causing substance was identified<br>• retrospectively, everything makes sense/understanding of what went wrong in the past<br>• being sure that skin reacts to mental health<br>• understanding that accumulated irritations/different factors led to eczema<br>• being aware of sensitive skin/abnormal skin<br>• many people are allergic to certain substances, it is normal to develop allergies<br>• understandable that skin reacts if it is not protected properly |
| | Example quote<br>"It's quite understandable. In so far as substances are hidden everywhere in many products, especially today, where many people suffer from allergic reactions anyway. So, it's not that I'm completely puzzled like: 'Does the good Lord want to punish me or something?' (Laughs.) Or something like-. I already have a good imagination of something like that." (B12) |

## Discussion

The aim of this qualitative study was to explore illness perceptions of occupational HE based on the dimensions of the common-sense model, including coherence and emotional representation. For this purpose, we conducted and analyzed guided face-to-face interviews with patients taking part in a three-week-inpatient rehabilitation program at a healthcare center specialized on occupational dermatology. The shortest of the interviews lasted 6 minutes. However, it delivered valuable information on the participants' individual illness perceptions. In this case, the interviewee was rather fact-oriented and delivered short-sentenced answers very quickly.

During analysis, several aspects became apparent which require further discussion. The great amount and variety of aspects named in some categories (i. e. external factors/irritant in 'causes', description of symptoms in 'label of symptoms and illness' as well as measures in 'controllability') had to be summarized under umbrella terms (see Tables 1–3). Even though we made sure to include all facets and formulations considered, some phenomena might not be shown in this paper.

Participants were diagnosed with different kinds of occupational HE and/or a combination of different diagnoses (e. g. ICD and AD). This might have led to varying results in some of the categories as discussed below. Furthermore, all participants of this study already attended at least two seminars on (occupational) skin diseases as part of the inpatient rehabilitation program and thus have gained some knowledge about, e. g. potential causes and protective measures. Hence, interviews with patients without any prior educational interventions might have led to different results.

'Causes perceived' is one of the biggest categories in our analysis. Most participants focused on external factors/irritants. Here, a great number of heterogeneous external factors was named. This might result from a wide range of substances used in different occupations which has already been discussed in a systematic review [43]. However, water (i. e. due to sweating under occlusive gloves, washing hands or working under wet conditions) was named by almost every single participant of the study. This is hardly surprising as wet work is considered as important risk factor for developing hand eczema [57]. Furthermore, some participants stated genetic factors, too, which might be due to different diagnoses (e. g., a combination of ICD and AD vs. patients which are only diagnosed with ICD). Besides, this study also shows that some patients believe in an abstract concept of 'tending to have allergies' which might describe atopic diseases such as AD. It is a pivotal finding of this paper, that due to combined diagnoses, patients with occupational HE identify different influential aspects of their disease. Lifestyle factors such as diet are also considered as potential causes in patients with occupational HE. This corresponds to Bathe et al. who also found nutrition as a factor mentioned by interviewees [35].

In accordance with a quantitative study [39], we found many participants describing the timeline of their occupational HE as periodical, cyclical or phasic. In addition to these findings, participants also illustrated the overall course of the disease as changing from phase to phase, either becoming better or worse. An important finding is that participants tended to 'tell' the course of their disease by describing changing symptoms. This might be due to the nature of the disease: In contrast to many other diseases, HE develops gradually and many symptoms are visually perceivable. Also, participants described the contexts under which they observed changes of their skin. This might also be relevant for discussing perceived causes of a disease or helpful measures. Even though participants were not asked directly, some interviewees stated that they are able to recognize slight skin changes at an early stage at the beginning of a phase. Hence, it can be assumed that some patients develop a kind of individual 'warning

system' alerting them when their skin commences deteriorating. Such 'warning system' seems to be an important aspect stimulating certain coping strategies and might be comparable to Leventhal et al.'s example of perceived symptoms which indicate high blood pressure [34]. Skin changes were similar at the beginning of each phase in a single participant. However, these markers differed amongst all interviewees. Even though this aspect is assigned to perception of timeline, as it was named in the context of the beginning of 'phases', this kind of 'warning system' might also have a great impact on the perceived controllability of the disease and should be investigated in further research. Furthermore, this might be a pivotal feature of (chronic) diseases with a cyclical timeline.

Studies on both occupational and non-occupational skin diseases state that these diseases are considered as highly symptomatic by the patients [43]. This is supported by our findings. Along with the categories cause and controllability, 'perceived symptoms' is one of the largest categories. Furthermore, participants did not only mention a great number, but also a great variety of different symptoms. Many of these focus on the visual appearance of the skin or changes in haptoral perception of its surface, but also unpleasant, non-visual sensations such as pruritus and pain. Many of the symptoms mentioned can be ascribed to all eczematous diseases. Furthermore, this study presents a variety of synonyms used by people with occupational HE to describe their illness. Not all of them deliver the image of an illness (such as "rash" or "inflammation") but rather a defective condition which needs to be adjusted (e. g. "bad skin" or "skin problem").

Considering controllability, this study rather emphasizes measures adopted by patients and/or physicians than a general concept of controllability. This might result from the question asked to cover this dimension ('What do you do when you notice your skin changing?'). Most certainly due to the nature of the disease, many participants unsurprisingly focus on the usage of different creams/ointments. However, this study also shows that self-harming procedures (e. g., opening vesicles) are sometimes considered to be a measure for relief. Some of these interviewees stated that they would not perform self-harming procedures anymore. This may be due to the interview situation itself and the fact that all participants were also patients who, in a medical context, are often expected to be compliant and perform a certain medically recommended behavior. Therefore, some respondents might have not even mentioned performing these procedures (social desirability bias). However, these procedures could be considered as a form of controllability which is achieved by covering the actual symptoms, e. g. replacing itchiness through pain. This should be taken into consideration in further studies as well as for educating and counseling patients.

In accordance with other studies [35, 36, 43], consequences perceived are primarily in the psychological, physical, private and occupational domains. Participants stated both implicitly and explicitly a perceived high burden of disease. This aspect is also confirmed in other studies [26–30, 43]. In this category, respondents tended to elaborate their personal experiences in particular detail which underlines the individual significance of these consequences in everyday life (see also [42]). This becomes particularly apparent through the sample quotes, but also through the summarized statements in Table 4.

The presented consequences of occupational HE affect all categories of the International Classification of Functioning, Disability and Health (ICF) [58]. Thus, interventions as well as the respective national structures of patient care should take all of these dimensions into account, e. g., by integrating interprofessional teams into care which not only include medical personnel, but also educational, psychological, and occupational therapists.

During analysis, some physical and psychological consequences were difficult to differentiate (e. g., sleeping disorder or limitations in everyday life). Also, distinguishing psychological consequences and emotional representation proved to be difficult (i. e., emotions names might

be considered as consequence, too). This has also been discussed in quantitative research using the CSM model [59]. Therefore, we decided to only code single words as sub-categories which describe a specific feeling in emotional representation, even though these might be understood as consequence as well. The category of psychological consequences on the other hand covers more complex statements on the perceived impact on the interviewee's mental health. Further studies–both quantitative and qualitative–using the CSM should be aware of this overlap.

Participants who elaborated aspects of coherence, aimed at identifying a specific cause (especially a positive patch test result) as a–desirable–guide to understand their disease. In this context, participants also expressed their wish to be able to manage their disease by knowing about the cause(s), which apparently is also an important factor in context of controllability. Both, the wish for knowing an actual cause and a tendency to use synonyms conveying the image of defective condition to describe the disease (see above), carry a strict biomedical understanding of health and illness [60], where a disease is emerging due to a (single) specific cause which needs to be identified and eliminated. This contradicts the complex disease pattern of occupational HE and should therefore be taken care of in communication between therapists and patients. Sloot et al. already suggested that patients should be educated about the multifactorial nature of HE [42]. This may also comprise a bio-psycho-social perspective and possible limitations of diagnostic tests at an early stage of the diagnostic phase.

Interestingly, a great number of participants considered their disease as both comprehensible and incomprehensible at the same time which might result from a combination of diagnoses (e. g., AD and ICD). In general, the use of the CSM in both qualitative and quantitative studies could yield different results in terms of coherence when dealing with diseases with different complexity of etiology. In terms of understanding coherence as a 'meta cognitive' level of the understanding of the disease [52], it might be beneficial to conduct further qualitative studies focusing on coherence only–in occupational HE or even AD, CD, ACD, and ICD separately. Furthermore, future (quantitative) studies should focus on deep analysis of differences and similarities in illness perceptions amongst patients with occupational HE (i. e. besides the form of HE mentioned above, also considering occupation, age), e. g. through aiming at a less heterogenous sample composition.

## Strengths and limitations of this study

To the best of our knowledge, this is the first study to use the CSM for a detailed investigation of illness perceptions in patients with occupational skin diseases. It delivers an in-depth description of the different dimensions and their complexity. Especially the dimensions timeline and coherence have not been investigated in depth before. A strength of this study is the usage of an open-ended stimulus at the beginning of the interview which encouraged interviewees to illustrate their individual experiences.

As all interviews were conducted in German language, example quotes and colloquial wording (see Tables 1–5) had to be translated into English language after analysis. Therefore, some of the example quotes presented might not comply with native colloquial English language.

The group investigated consists of inpatients of a healthcare center specialized on occupational dermatology in Germany. Due to these very specific arrangements, the transferability of our results to other patient groups (e. g. with milder forms of occupational HE) might be limited. Furthermore, as all interviewees were patients of this healthcare center and both interviewers work as health educators, social desirability might have biased the participants' statements during the interviews.

## Conclusion

The aim of this qualitative study was to explore the illness perceptions of patients with occupational HE. The qualitative approach allowed participants to express their experiences and perceptions in great detail. As our results show, patients use a variety of terms and expressions to convey their illness perceptions. The great amount of causes named by the participants often derive from their occupational life: Besides external irritants and other hazardous factors as well as allergies, psycho-social factors appear to be central perceived causes. Also, the impact of occupational HE on the individual's life is highlighted by the wide range of consequences affecting all areas of life. These aspects highlight the importance of psycho-social counselling of patients with occupational HE. All in all, knowledge on illness perceptions of patients with occupational HE is crucial for communication between healthcare professionals and patients. Especially since the individual behavior and self-management of the disease are of great importance for treating the disease. In regard to applying the CSM, this study shows that this model appears to be appropriate to investigate illness perceptions of occupational HE. However, this paper highlights some challenges which may arise when working with the CSM–both in quantitative and qualitative studies. First, these challenges make it necessary to reflect on the nature of the disease and its etiology. Further qualitative and quantitative studies on illness perceptions should be sensitive to the potential effect on results when interviewing a group of patients who have slightly varying medical diagnoses as well as different causes and triggers of the disease (i. e., illnesses evolving from genetic disposition vs. lifestyle). Also, this study shows the difficulty of differentiating between emotional representation and psychological consequences. The complexity of illness perceptions of occupational HE presented in this study underlines the multidimensionality of diseases such as occupational HE. Hence, subjective illness perceptions should be reflected by professionals of all disciplines involved in working with those patients (i. e. should be considered in patient training programs and counseling).

## Supporting information

**S1 Appendix. Standards for Reporting Qualitative Research (SRQR).**
(DOCX)

**S2 Appendix. Interview guide.**
(DOCX)

**S3 Appendix. Category system.**
(DOCX)

**S4 Appendix. Original German quotes and English translations as presented in publication.**
(DOCX)

## Acknowledgments

We would like to thank all participants in this study for contributing their time and personal experiences to this study.

## Author Contributions

**Conceptualization:** Anna-Sophie Buse, Swen Malte John, Andreas Hansen.

**Data curation:** Anna-Sophie Buse, Andreas Hansen.

**Formal analysis:** Anna-Sophie Buse, Andreas Hansen.

**Funding acquisition:** Annika Wilke, Swen Malte John.

**Investigation:** Anna-Sophie Buse, Andreas Hansen.

**Methodology:** Anna-Sophie Buse, Andreas Hansen.

**Project administration:** Annika Wilke, Swen Malte John.

**Resources:** Swen Malte John.

**Supervision:** Annika Wilke, Swen Malte John, Andreas Hansen.

**Writing – original draft:** Anna-Sophie Buse.

**Writing – review & editing:** Annika Wilke, Swen Malte John, Andreas Hansen.

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
