## [Decision Letter · Decision Letter 0]

27 Mar 2023

PONE-D-22-34217Illness perceptions of occupational hand eczema in German patients based on the common-sense model of self-regulation: A qualitative studyPLOS ONE

Dear Dr. Buse,

Thank you for submitting your manuscript to PLOS ONE. After careful consideration, we feel that it has merit but does not fully meet PLOS ONE’s publication criteria as it currently stands. Therefore, we invite you to submit a revised version of the manuscript that addresses the points raised during the review process.

We look forward to receiving your revised manuscript.

Kind regards,

Aiggan Tamene

Academic Editor

PLOS ONE

Journal Requirements:

“We acknowledge support by Deutsche Forschungsgemeinschaft (DFG) and Open Access Publishing Fund of Osnabrück University. The project ‘Mixed-methods study to assess illness perceptions of patients with occupational contact dermatitis of the hands for enhancing patient education and counseling’ [‘SubjeKt: Mixed-Methods-Studie zur Erfassung subjektiver Krankheitstheorien von Patientinnen und Patienten mit berufsbedingten Handekzemen für die Schulungs- und Beratungspraxis’], project no. ext FF_1436, was funded by the Institution for Statutory Accident Insurance and Prevention in the Health and Welfare Services [Berufsgenossenschaft für Gesundheitsdienst und Wohlfahrtspflege]. The funding institution was not involved in the study design, analysis and interpretation of the data, writing the manuscript or the decision to submit the manuscript for publication.”

“The project ‘Mixed-methods study to assess illness perceptions of patients with occupational contact dermatitis of the hands for enhancing patient education and counseling’ [‘SubjeKt: Mixed-Methods-Studie zur Erfassung subjektiver Krankheitstheorien von Patientinnen und Patienten mit berufsbedingten Handekzemen für die Schulungs- und Beratungspraxis’], project no. ext FF_1436, was funded by the Institution for Statutory Accident Insurance and Prevention in the Health and Welfare Services [Berufsgenossenschaft für Gesundheitsdienst und Wohlfahrtspflege]. Furthermore, the publication of this study was supported by Deutsche Forschungsgemeinschaft (DFG) and Open Access Publishing Fund of Osnabrück University.

Additional Editor Comments:

Thank you for submitting your manuscript, to PLOS ONE. After careful consideration, it appears that there are some modifications that will need to be made before we are able to accept it for publication. Your work is a first-rate example of the genre and continues to provide an in-depth examination of the topic at hand. The organized structure, complemented by compelling evidence, results in an engaging and informative read. Its insights are invaluable and will contribute richly to the conversation. Having said this, I still believe that the main argument of your submission could be further fleshed out and strengthened with additional evidence and analysis. Additionally, citations throughout the article should adhere to accepted scholarly standards in PLOS ONE.

Reviewers' comments:

Reviewer's Responses to Questions

**Comments to the Author**

1. Is the manuscript technically sound, and do the data support the conclusions?

Reviewer #1: No

Reviewer #2: Yes

2. Has the statistical analysis been performed appropriately and rigorously? 

Reviewer #1: N/A

Reviewer #2: N/A

3. Have the authors made all data underlying the findings in their manuscript fully available?

Reviewer #1: No

Reviewer #2: No

4. Is the manuscript presented in an intelligible fashion and written in standard English?

Reviewer #1: Yes

Reviewer #2: Yes

5. Review Comments to the Author

Reviewer #1: Thank you authors for conducting your research on interesting title “Illness perceptions of occupational hand eczema in German 4 patients based on the common-sense model of self-regulation: A qualitative study”. It would benefit for taking action to improve occupational health for work safety. Please find below my comments to improve quality of the paper before publication. Abstract: Which qualitative study design was applied in this study? It is not clear how you analyzed the data. Please provide more description, including any software used to manage data analysis. On the method section you mentioned thirty sex individuals were interviewed, but on the result section, thirty five transcripts were transcribed verbatim and analyzed. Why? The first two sentences of the result part need to be written under method part. Things written under discussion section are not consistent with what was written under method and result part. Why it is needed further qualitative study once you included all dimensions of the model? Introduction: Well-written! Methods Study design: Not clear. Which study design you employed? Recruitment: Statements written here contained concepts that would be incorporated under ethical considerations. Rather write, in detail, about the sampling technique (s) and criteria’s considered to recruit participates. Merge sampling and recruitment together and describe in detail. Data collection: It is surprising that you conducted a 6 minute interview. How did you see it? What is the minimum interview time? You mentioned that the guide was prepared in relation to the dimensions of the model. So, how you see it. This needs especial attention. Additionally, how you fixed the sample size? Were participants come from one organization of different? This is because; different organizations might have different occupational health hazards, which in turn, vary your sample size to reach saturation. Trustworthiness: Your paper lacks trustworthiness section, which is one of the most important sections in qualitative research report. Result: Please provide detail socio-demographic characteristics of the participants. Please provide quote (s) to each categories or themes. Discussion: Please provide implication or interpretations for the findings. ConclusionsPlease rewrite the conclusion section to be consistent with your research findings. Write what you understood from the findings.

Reviewer #2: The manuscript is well written and is technically sound. The methodology employed is acceptable, and the results, discussion and conclusion are well thought through and can make a positive contribution to literature and knowledge. The authors have indicated that the data cannot be made available as the study participants did not consent to that. It is particularly good that the strengths and limitations of the study are clearly discussed.

6. PLOS authors have the option to publish the peer review history of their article (what does this mean?). If published, this will include your full peer review and any attached files.

Reviewer #1: No

Reviewer #2: No

---

## [Author Response · Author response to Decision Letter 0]

21 Apr 2023

Point-by-point response for the revision of the manuscript “Illness perceptions of occupational hand eczema in German patients based on the common-sense model of self-regulation: A qualitative study”

Editor Comments

Thank you for submitting your manuscript, to PLOS ONE. After careful consideration, it appears that there are some modifications that will need to be made before we are able to accept it for publication. Your work is a first-rate example of the genre and continues to provide an in-depth examination of the topic at hand. The organized structure, complemented by compelling evidence, results in an engaging and informative read. Its insights are invaluable and will contribute richly to the conversation. Having said this, I still believe that the main argument of your submission could be further fleshed out and strengthened with additional evidence and analysis. Additionally, citations throughout the article should adhere to accepted scholarly standards in PLOS ONE.

Response: We thank the editor for reading our manuscript and the comments. We excluded direct quotes from the manuscript body in order to adhere citation standards of ‘Vancouver style’ (see lines 84-87 and 150-154). Thank you for your advice that the main argument of our manuscript could be further fleshed out. This was also suggested by Reviewer 1. Based on your feedback, we added a more precise description of our findings and its implications for practice to our conclusion. However, the aim of our study was to qualitatively explore and present illness perceptions of patients with occupational HE in general. In line with this study aim, we believe that analyses conducted so far allow a first, broad insight into these illness perceptions and we are concerned that further analysis might significantly and unfavorably increase the length of the paper. Yet, we strongly suggest (see also line 418-421) that further research should lay a special focus on the influence of sociodemographic factors on illness perceptions of patients with occupational HE (e. g. regarding different workplaces). 

Journal Requirements

Point 1:

Response 1: We have carefully re-checked the style requirements and renamed the attachment files. 

Point 2: Thank you for stating the following in the Acknowledgments Section of your manuscript:

“We acknowledge support by Deutsche Forschungsgemeinschaft (DFG) and Open Access Publishing Fund of Osnabrück University. The project ‘Mixed-methods study to assess illness perceptions of patients with occupational contact dermatitis of the hands for enhancing patient education and counseling’ [‘SubjeKt: Mixed-Methods-Studie zur Erfassung subjektiver Krankheitstheorien von Patientinnen und Patienten mit berufsbedingten Handekzemen für die Schulungs- und Beratungspraxis’], project no. ext FF_1436, was funded by the Institution for Statutory Accident Insurance and Prevention in the Health and Welfare Services [Berufsgenossenschaft für Gesundheitsdienst und Wohlfahrtspflege]. The funding institution was not involved in the study design, analysis and interpretation of the data, writing the manuscript or the decision to submit the manuscript for publication.”

“The project ‘Mixed-methods study to assess illness perceptions of patients with occupational contact dermatitis of the hands for enhancing patient education and counseling’ [‘SubjeKt: Mixed-Methods-Studie zur Erfassung subjektiver Krankheitstheorien von Patientinnen und Patienten mit berufsbedingten Handekzemen für die Schulungs- und Beratungspraxis’], project no. ext FF_1436, was funded by the Institution for Statutory Accident Insurance and Prevention in the Health and Welfare Services [Berufsgenossenschaft für Gesundheitsdienst und Wohlfahrtspflege]. Furthermore, the publication of this study was supported by Deutsche Forschungsgemeinschaft (DFG) and Open Access Publishing Fund of Osnabrück University.

Response 2:

Thank you very much for bringing this point to our attention. We removed any funding-related text from the manuscript. We included the amended statements in the cover letter.

Point 3: We note that the grant information you provided in the ‘Funding Information’ and ‘Financial Disclosure’ sections do not match.

Response 3: We added the project number (ext FF_1436) of the BGW project to the Funding Information section. There is no number for the funding by the DFG (Deutsche Forschungsgemeinschaft). 

Point 4: We note that you have indicated that data from this study are available upon request. PLOS only allows data to be available upon request if there are legal or ethical restrictions on sharing data publicly. For more information on unacceptable data access restrictions, please see http://journals.plos.org/plosone/s/data-availability#loc-unacceptable-data-access-restrictions.

Response 4: Thank you for your comment. We changed our data availability statement accordingly and included it into the cover letter. 

Point 5: Please review your reference list to ensure that it is complete and correct. If you have cited papers that have been retracted, please include the rationale for doing so in the manuscript text, or remove these references and replace them with relevant current references. Any changes to the reference list should be mentioned in the rebuttal letter that accompanies your revised manuscript. If you need to cite a retracted article, indicate the article’s retracted status in the References list and also include a citation and full reference for the retraction notice.

Response 5: Thank you for your comment. We have carefully re-checked the reference list and can assure you that we have not included any papers that have been retracted.

 

Response to Reviewer 1

Point 1: Thank you authors for conducting your research on interesting title “Illness perceptions of occupational hand eczema in German 4 patients based on the common-sense model of self-regulation: A qualitative study”. It would benefit for taking action to improve occupational health for work safety. Please find below my comments to improve quality of the paper before publication.

Response 1: We thank the reviewer for carefully reading our manuscript. We have intensively discussed the comments and revised the manuscript according to this feedback.

Point 2 (referring to Abstract): Which qualitative study design was applied in this study? 

Response 2: Thank you very much for your comment. We chose an exploratory qualitative approach in our study. We added the term “exploratory” in our Abstract (line 31). Considering data analysis, we applied a qualitative text analysis, which is mentioned in the abstract (line 36-37). In our Methods section, we provide further information. Here, we also refer to the publication by Kuckartz (“Qualitative text analysis: A guide to methods, practice & using software “) [56] which thoroughly describes this approach.

Point 3 (referring to Abstract): It is not clear how you analyzed the data. Please provide more description, including any software used to manage data analysis. 

Response 3: Thank you very much for bringing this to our attention. We have added this information into the abstract (line 36-38):

Data was analyzed deductively and inductively using qualitative text analysis. MAXQDA 2018 (Verbi, Berlin, Germany), a software for qualitative data analysis, was applied for coding and summarizing of results.

Point 4 (referring to Abstract): On the method section you mentioned thirty sex individuals were interviewed, but on the result section, thirty five transcripts were transcribed verbatim and analyzed. Why? 

Response 4: In our results section, we described why one of the interviews was excluded from analysis afterwards. We agree that this should also be described in the abstract and hence, added the following explanation in line 34-36:

One interview had to be excluded before analysis, since one participant’s diagnosis was retrospectively changed from ICD to tinea and hence did not match the inclusion criteria. Thirty-five interviews were transcribed verbatim and analyzed.

To improve readability and understanding throughout the manuscript, we removed the explanation from the results section and added respective notes into the data collection and data analysis sections (see tracked changes within the manuscript).

Point 5 (referring to Abstract): The first two sentences of the result part need to be written under method part. 

Response 5: Thank you very much to your comment. We changed the abstract accordingly (see tracked changes in lines 36 and 39-40).

Point 6 (referring to Abstract): Things written under discussion section are not consistent with what was written under method and result part. Why it is needed further qualitative study once you included all dimensions of the model? 

Response 6: Thank you very much for pointing this out. We suggest that future qualitative studies should only look at coherence, as it is a very complex construct. Hence, it would be interesting to see how coherence is represented in different disease patterns that have different complexity. In our opinion, a qualitative approach would be a great advantage to analyze these aspects. We understand that the wording in our abstract is misleading. Therefore, based on your feedback, we changed the sentence as follows (line 52-54):

Future research might further investigate specific aspects of illness perceptions in patients with occupational HE, especially considering the complexity of coherence and overlapping dimensions (i. e. emotional representation and psychological consequences). 

Point 6 (referring to Methods): Study design: Not clear. Which study design you employed? 

Response 7: Thank you for your comment. As mentioned under point 2 of this revision, we conducted an exploratory qualitative approach. We added the term “exploratory” in our Methods section (see tracked changes, line 117). 

Point 7 (referring to Methods): Recruitment: Statements written here contained concepts that would be incorporated under ethical considerations. Rather write, in detail, about the sampling technique (s) and criteria’s considered to recruit participates. Merge sampling and recruitment together and describe in detail. 

Response 7: Thank you very much for pointing this out. We agree that we lay a special focus on ethical aspects in our recruitment section which is a sub-section of methods. However, we consider ethical aspects as important information due to the fact that we worked with patients. According to your feedback, we merged Recruitment and Sample (see tracked changes in the manuscript, line 122). Furthermore, we included a more detailed description of our sample (line 134-141).

Point 8 (referring to Methods): Data collection: It is surprising that you conducted a 6 minute interview. How did you see it? What is the minimum interview time? You mentioned that the guide was prepared in relation to the dimensions of the model. So, how you see it. This needs especial attention.

Response 8: Thank you very much for your comment. We agree, that one of the interviews was rather short. However, this was the shortest interview in our data collection (average length was approx. 16 minutes) and despite its shortness, it delivered valuable information on the participants’ individual illness perceptions. In this case, the interviewee was rather fact-oriented and delivered short-sentenced and fact-oriented answers very quickly. The interviewee was neither shy or reticent. Against the background of your feedback, we included the mean interview time into our data collection section (see line 157-158). Furthermore, we added this aspect into our discussion (line 312-314): 

The shortest of the interviews lasted 6 minutes. However, it delivered valuable information on the participants’ individual illness perceptions. In this case, the interviewee was rather fact-oriented and delivered short-sentenced and fact-oriented answers very quickly. 

Point 9 (referring to Methods): Additionally, how you fixed the sample size? Were participants come from one organization of different? This is because; different organizations might have different occupational health hazards, which in turn, vary your sample size to reach saturation. 

Response 9: Thank you very much for your comment. We agree that different occupations and different workplaces (even if the respective occupation is the same) lead to different health threads/skin hazards. However, this was not the focus of our study, especially due to the very heterogenous sample. In accordance to your feedback (see also point 7) we included a more detailed description of our sample (line 134-141) and suggest further research to this aspect in our discussion (see line 418-421).

Point 10 (referring to Methods): Trustworthiness: Your paper lacks trustworthiness section, which is one of the most important sections in qualitative research report. 

Response 10: Thank you for pointing this out. We already included a completed SRQR checklist as appendix to follow reporting standards for qualitative research which we note under Methods. This is a standard tool in qualitative research papers. However, we are willing to revise our manuscript if the editor of the journal advises us to do so.

Point 11 (referring to Results):

Please provide detail socio-demographic characteristics of the participants. 

Response 11: Thank you very much for your comment. It would definitely be interesting to take a thorough look on the influence of socio-demographic characteristics on illness perceptions of patients with occupational HE. However, this was not the aim of this paper. Nevertheless, based on your feedback, we included further details on our sample in the Sample section (see also point 7 of this revision). 

Point 12 (referring to Results):

Please provide quote (s) to each categories or themes. 

Response 12: Thank you very much for your comment. We agree that quotes in every category are useful and added them under the categories ‘timeline’ and ‘emotional representation’ which were lacking quotes. 

Point 13 (referring to Discussion): 

Please provide implication or interpretations for the findings. 

Response 13: Thank you for pointing this out. As stated in our discussion, we find it important to consider illness perceptions of patients with occupational HE as a pivotal part of communication between patients and therapists (i. e. doctors, health educators, etc.). We also added this aspect to our conclusion in order to deliver a more precise understanding of our findings (see point 14).

Point 14 (referring to Conclusion):

Please rewrite the conclusion section to be consistent with your research findings. Write what you understood from the findings. 

Response 14: Thank you very much for your comment which we have intensely discussed. We have added more detailed information on our findings and its implications for practice (see tracked changes in lines 441-446 and 447-449). 

Response to Reviewer 2

Point 1: The manuscript is well written and is technically sound. The methodology employed is acceptable, and the results, discussion and conclusion are well thought through and can make a positive contribution to literature and knowledge. The authors have indicated that the data cannot be made available as the study participants did not consent to that. It is particularly good that the strengths and limitations of the study are clearly discussed.

Response 1: We thank the reviewer for the careful reading of the manuscript and the positive response.

---

## [Editor Report · Decision Letter 1]

2 May 2023

Illness perceptions of occupational hand eczema in German patients based on the common-sense model of self-regulation: A qualitative study

PONE-D-22-34217R1

Dear Dr. Buse,

We’re pleased to inform you that your manuscript has been judged scientifically suitable for publication and will be formally accepted for publication once it meets all outstanding technical requirements.

Kind regards,

Aiggan Tamene

Academic Editor

PLOS ONE
---

## [Editor Report · Acceptance letter]

5 May 2023

PONE-D-22-34217R1 

Illness perceptions of occupational hand eczema in German patients based on the common-sense model of self-regulation: A qualitative study 

Dear Dr. Buse:

I'm pleased to inform you that your manuscript has been deemed suitable for publication in PLOS ONE. Congratulations! Your manuscript is now with our production department. 

Kind regards, 

on behalf of

Mr Aiggan Tamene 

Academic Editor

PLOS ONE